# HyperGrid Transformers: Towards A Single Model for Multiple Tasks

**Yi Tay, Zhe Zhao, Dara Bahri, Donald Metzler, Da-Cheng Juan**
Google Research
Mountain View, California
{`yitay,zhezhao,dbahri,metzler,dacheng`}@google.com

## Abstract

Achieving state-of-the-art performance on natural language understanding tasks typically relies on fine-tuning a fresh model for every task. Consequently, this approach leads to a higher overall parameter cost, along with higher technical maintenance for serving multiple models. Learning a single multi-task model that is able to do well for all the tasks has been a challenging and yet attractive proposition. In this paper, we propose HyperGrid Transformers, a new Transformer architecture that leverages task-conditioned hyper networks for controlling its feed-forward layers. Specifically, we propose a decomposable hypernetwork that learns grid-wise projections that help to specialize regions in weight matrices for different tasks. In order to construct the proposed hypernetwork, our method learns the interactions and composition between a global (task-agnostic) state and a local task-specific state. We conduct an extensive set of experiments on GLUE/SuperGLUE. On the SuperGLUE test set, we match the performance of the state-of-the-art while being 16 times more parameter efficient. Our method helps bridge the gap between fine-tuning and multi-task learning approaches.

## 1 Introduction

Learning a single multi-task model that performs well across multiple targeted tasks is an attractive proposition for many reasons (Kaiser et al., 2017; Ruder, 2017; Clark et al., 2019b). Although extremely challenging, this paradigm enables a substantial savings in overall parameter costs, along with eliminating the need for maintaining multiple models in production (Stickland and Murray, 2019). However, achieving state-of-the-art performance on natural language understanding benchmarks today (Wang et al., 2018; 2019) still relies on fine-tuning a new model for every single task. This methodology is infeasible in many situations. Moreover, certain tasks rely on an extensive ensemble of models and/or task-specific fine-tuning tricks (Liu et al., 2019b; Devlin et al., 2018; Clark et al., 2020).

The single-task fine-tuning paradigm is well-established to be the dominant approach (Raffel et al., 2019), as training multiple tasks using a single set of parameters can be problematic in many ways, such as catastrophic forgetting (French and Chater, 2002; McCloskey and Cohen, 1989; McClelland et al., 1995; Kirkpatrick et al., 2017) or the inherent difficulty of finding a consistently good model for all tasks (Clark et al., 2019b; Wu et al., 2020). Inevitable task conflicts and difficulty in fitting all models within a set of hard parameters is also a challenging problem for multi-task co-training.

In this paper, we propose a new Transformer architecture, the HyperGrid Transformer for efficient modeling of multiple tasks within a single set of model parameters. HyperGrid Transformers rely on a hypernetwork-based (Ha et al., 2016) module that performs *gridwise decomposable hyper projections*. This module is task conditioned and dynamically learns to generate weights of the feed-forward layers of the Transformer model.

Overall, our eventual goal is to dispense with task specific fine-tuning tricks altogether. While neural networks typically maintain the same consistent set of parameters for all input instances, the proposed HyperGrid Transformers introduces instance-specific parameters by conditioning on the current input. This setup enables our model to learn task-specific reparameterization for each input instance, which mitigates several challenges of multi-task co-training.

Our proposed HyperGrid module belongs to a family of hypernetworks (Ha et al., 2016), in which a side network is responsible for weight generation for the main network. In our case, task-conditioned hypernetworks provide greater flexibility and expressiveness for capturing the dynamics of multiple tasks within a single set of parameters. Specifically, we introduce two novel algorithmic improvements over the existing methods.

First, we introduce the notion of grid-wise projections in which we assume a structural layout in vanilla projection layers. For each input sample, our projections dynamically control the parameters in a grid-wise, region-specific manner. The structural segmentation of feed-forward layers is similar in spirit to mixture-of-experts gating (Shazeer et al., 2017), albeit at a lower-level. Conversely, standard hypernetworks only consider row-wise re-weighting of weight matrices.

Second, we introduce *decomposable* hyper-projections. The key idea is to learn rich compositional and pairwise interactions between dual hypernetworks. A dual setup is adopted, where we explore different hypernetwork composition variants. We introduce a novel local-global setup, which composes a local instance-specific and task-specific hyper-projection with a task agnostic global state embedding. This is intuitive since this setup is not only highly expressive and flexible but also serves as a factorization of local and global components. To the best of our knowledge, our work is the first to explore this setup with respect to learning conditional parameters.

Finally, we conduct extensive experiments on GLUE/SuperGLUE. Our proposed model is able to match the performance of individually fine-tuned state-of-the-art Text-to-Text Transformers (T5) (Raffel et al., 2019) models with a *single* model that is learned to fit all GLUE and SuperGLUE tasks at once. Moreover, our single model also outperforms strong baselines that employ ensembling and other task-specific tricks (Liu et al., 2019b; Clark et al., 2020).

**Our Contributions**    The contributions of this paper can be summarized as follows:

- We propose HyperGrid Transformers, a form of hypernetwork-based Transformer that learns task-conditioned dynamic weights for its feed-forward layers.
- The key novelty behind HyperGrid Transformers is the factorization of local and global components for weight generation. Our weight generation is grid-wise and imbues the model with a structural layout.
- We conduct extensive experiments on natural language understanding benchmarks (GLUE/SuperGLUE). With a single model, we match the state-of-the-art T5 model that is finetuned in a per-task fashion (multiple models), resulting in 16x parameter savings.

## 2    HYPERGRID TRANSFORMERS

This section outlines the key idea of the proposed algorithm.

### 2.1    HYPERGRID MODULE

HyperGrid operates on weight matrices (linear transformations), i.e., $Y = \boldsymbol{W}X + b$. In a hypernetwork formulation, instead of letting $\boldsymbol{W}$ be free weights, we generate $\boldsymbol{W}$ using a parameterized side network $H(.)$.

$$Y = \boldsymbol{W}x + b \quad \text{where} \quad \boldsymbol{W} = H(X) \tag{1}$$

where $\boldsymbol{W} \in \mathbb{R}^{d_m \times d_f}$. In the case where $X$ is a single vector $\in \mathbb{R}^{d_m}$, we may parameterize $H(.)$ with a simple feed-forward layer.

$$H(X) = \sigma(\boldsymbol{U}X)\mathbf{1}^{\top} \odot \boldsymbol{W} \tag{2}$$

where $\mathbf{1}$ is a column vector of ones, $\sigma$ is the sigmoid activation function and $U \in \mathbb{R}^{d_m \times d_f}$. The key idea is that the hypernetwork generates a vector, i.e., $\boldsymbol{U}X \in \mathbb{R}^{d_f}$ that is broadcast (multiplied by $\mathbf{1}$) and multiplied by $\boldsymbol{W}$, acting as a row-wise scaling of $\boldsymbol{W}$. We are also able to reduce $U \in \mathbb{R}^{d_m \times n}$ where $d_f \mod n = 0$ and repeat the vector $\frac{d_f}{n}$ times to form the original dimension of $d_f$. These methods only consider scaling one dimension of $W$ (e.g., row-wise). We now consider methods beyond simple row-wise weight scaling.

### 2.1.1 Decomposable Gridwise Projections

In our method, we propose grid-wise projections that segments $W$ into a grid, i.e., blocks of $\frac{d_m}{d_r} \times \frac{d_f}{d_c}$. We generate blocks by the outer product of $L_r \in \mathbb{R}^{d_r}$ and $L_c \in \mathbb{R}^{d_c}$. Note that $d_r$ and $d_c$ are user-specified hyperparameters that control the grid-size for the fan-in and fan-out of the output matrix. For simplicity, we consider divisible blocks where $d_r < d_m, d_m \mod d_r = 0$, and $d_c < d_f, d_f \mod d_c = 0$. In this case:

$$H(X) = \psi(\sigma((L_r X)(L_c X)^\top)) \odot W \quad (3)$$

where $(L_r X)(L_c X)^\top \in \mathbb{R}^{d_r \times d_c}$, $\psi(.)$ is a repeat vector function that repeats its input $\frac{d_m}{d_r}$ times on the row axis and $\frac{d_f}{d_c}$ times on the column axis. We name this approach the $L^2$ variant, short for Local-Local Gridwise Projection.

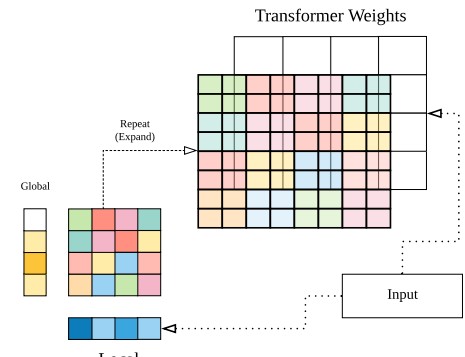

Figure 1: Detailed Illustration of the proposed Decomposable Gridwise Projections. Two decomposable vectors compose to form a gating matrix which is expanded to construct task-adaptive weight matrices.

**Composition between Local and Global Factors** The decomposable grid-wise projections learn $L_r$ and $L_c$ from $X$, which makes it conditioned on local, instance-wise information. Here, we postulate that it may be beneficial for either $L_r$ or $L_c$ to be a global embedding. By keeping $L_c$ as a global, trainable embedding, this can be formulated as:

$$H(X) = \psi(\sigma((L_r X) G_c^\top)) \odot W \quad (4)$$

where $G_c \in \mathbb{R}^{d_f}$. In this case, $L_r$ is conditioned from $X$, the specific input sample. On the other hand, $G_c$ remains consistent across all input samples. Hence, the outer product is essentially a rich dyadic composition between local and global factors.

**Local-Global and Global-Local** It is easy to see that there are two ways of composing $L$ and $G$. The above method considers the Local-Global approach where the fan-in uses a local hypernetwork and the global part uses a trainable embedding. An alternative that flips this around to use a Global-Local composition is evaluated in our experiments. Namely, this can be expressed as:

$$H(X) = \psi(\sigma((G_r(L_c X)^\top)) \odot W \quad (5)$$

### 2.2 Dynamic Weight Generation with HyperGrid

This section describes how we use task-conditioned hypernetworks to influence and generate the parameters for HyperGrid Transformers.

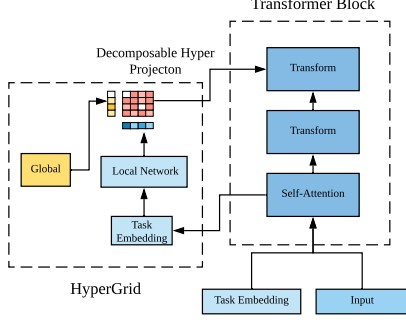

Figure 2: Illustration of the proposed Hyper-Grid architecture.

**Task Conditioning** The local network part of Hyper-Grid $L$ is learned via a task embedding $T \in \mathbb{R}^{d_m}$, which provides a task identifier and information to the hypernetwork. In HyperGrid Transformers, we first apply self-attention of the task embedding by concatenating it with the input sequence. This is described as:

$$T' = \text{MHSA}([T; X])_{[0]} \quad (6)$$

where $[;]$ is a concatenation on the length dimension and MHSA(.) is the multi-head self-attention function. The input sequence $X$ interacts with the task embedding to generate $T'$ which is used in our hypernetwork module.

**Weight Gating**  The HyperGrid module is added at the position-wise feed-forward layers of the Transformer models. More specifically, we equip the second positional FFN after the ReLU activations with HyperGrid. There are several reasons for doing so. In most Transformer implementations, the fan out of this layer is typically scaled up to very large values (Raffel et al., 2019). Hence, the influence on this layer has the greatest potential to benefit the Transformer model. Second, early experiments on both of the positional feed-forward layers yielded no substantial improvements. Hence, we opt to only modify the second positional FFN of the Transformer model.

**Initialization**  In our experiments, we take advantage of existing pretrained model (Raffel et al., 2019) and add additional HyperGrid parameters that are fine-tuned along with the rest of the network. The overall formulation of the HyperGrid-enhanced Transformer can be written as:

$$Y_i = H_i(X_{i-1}, \boldsymbol{W_i}) + \boldsymbol{W_i}(X_{i-1}) \tag{7}$$

where $i$ denotes the layer $i$. We construct a new HyperGrid (with non-shared parameters) for each layer. Since $\boldsymbol{W}$ has been pretrained, we also add a residual connection of the original $\boldsymbol{W_i}(X_{i-1})$ computation to the mixture.

**Parameter Costs**  We note that the parameter counts added by HyperGrid are relatively negligible since $d_r$ and $d_c$ are small. In the $LG$ setting, the model adds $d_m d_r + d_c$ parameters at each layer. On the $GL$ setting, the parameter cost added is $d_r + d_f d_c$. The most expensive option is $L^2$ where the added cost is $d_m d_r + d_f d_c$. These added parameter costs are often negligible for large Transformer models.

# 3  EXPERIMENTAL RESULTS

We conduct experiments on GLUE (Wang et al., 2018) and SuperGLUE (Wang et al., 2019) which are consolidated benchmarks of multiple challenging NLP and NLU tasks. While most of the work in this area has been focused on achieving good task-specific performance, our work focuses on trying to get good performance with a single model on **all** GLUE and SuperGLUE tasks. Therefore, most of our experiments are conducted on a mixture of all GLUE and SuperGLUE tasks.

## 3.1  EXPERIMENTAL RESULTS

In this section, we discuss the empirical results of our experiments. Further details about the experimental setup can be found in the appendix. Pertaining to parameter counts (reported as $\theta$ in our experiments), this is the **total** parameter cost for serving the entire suite of tasks. If a single model (finetuned on a single task) is $X$ parameters, then the costs of serving $N$ tasks in a multi-model setup would be $NX$.

### 3.1.1  SINGLE MODEL VERSUS MULTIPLE MODELS

These experiments investigate different model settings, namely using (1) multiple models (MM) or a single model (SM) for all tasks. The MM model trains a fresh model on every single task and retains the best checkpoint for all tasks. Meanwhile, the SM model is trained on all tasks at once. We compare them against HGT, our *single model* HyperGrid Transformer approach.

**Details about Baselines**  For the Single Model Baseline, this is the identical T5 model without our HyperGrid layers which serves as a fair comparison to our model. This enables us to observe the effect of HyperGrid directly. As for the multi-model method, this is done by finetuning T5 directly on each task and reporting results from the best checkpoint (of each task). This is a very strong baseline because the model is also allowed to choose specific best checkpoints for each task which the single model approach is prohibited against. As for the sampling strategy in the single-model approach (baseline and our approaches), we use a proportionate mix of tasks according to the number of samples in the each task/dataset.

| Model | $|\theta|$ | Avg | CoLA | SST | MR | STS | QQP | MNLI | QNLI | RTE |
|---|---|---|---|---|---|---|---|---|---|---|
| T5[†] (SOTA) | 3.2B | 83.4 | 53.8 | 92.7 | 88.9 | 88.0 | 91.6 | 84.4 | 90.5 | 76.3 |
| Multi Model | 3.2B | 85.7 | 59.6 | 94.2 | 90.1 | 89.1 | 90.6 | 86.5 | 93.7 | 82.0 |
| Single Model | 0.2B | 85.0 | 57.3 | 94.2 | 88.6 | 89.5 | 90.2 | 86.2 | 93.1 | 80.9 |
| **Our Single-Model Approaches** | | | | | | | | | | |
| HGT ($L^2$) | 0.2B | 85.2 | 59.4 | 90.6 | 90.1 | 88.9 | 90.3 | 86.5 | 93.1 | 79.1 |
| HGT ($LG$) | 0.2B | 85.4 | 57.9 | 94.6 | 89.2 | 90.1 | 90.3 | 86.7 | 81.2 | 84.2 |
| HGT ($L$) | 0.2B | 85.6 | 59.9 | 94.0 | 89.1 | 89.9 | 90.2 | 86.5 | 93.1 | 81.1 |

Table 1: Experimental results on GLUE dev set.

| Model | $|\theta|$ | Avg | BQ | CB | CP | MultiRC | Record | RTE | WiC | WSC |
|---|---|---|---|---|---|---|---|---|---|---|
| T5[†] (SOTA) | 3.2B | 71.4 | 76.6 | 91.2/92.0 | 66.2 | 66.1/25.8 | 69.1/68.2 | 75.3 | 68.0 | 78.6 |
| Multi Model | 3.2B | 74.8 | 82.9 | 96.4/92.0 | 63.0 | 79.1/44.0 | 77.6/76.8 | 83.8 | 71.6 | 73.1 |
| Single Model | 0.2B | 73.6 | 81.5 | 77.3/83.9 | 64.0 | 78.2/43.3 | 76.9/76.1 | 84.1 | 66.9 | 74.0 |
| **Our Single-Model Approaches** | | | | | | | | | | |
| HGT ($L^2$) | 0.2B | 75.3 | 82.4 | 85.3/91.1 | 64.0 | 77.8/42.7 | 76.8/75.9 | 83.4 | 67.1 | 80.8 |
| HGT ($LG$) | 0.2B | 74.8 | 82.5 | 83.1/89.3 | 64.0 | 77.9/42.8 | 77.1/76.3 | 84.1 | 65.5 | 78.8 |
| HGT ($L$) | 0.2B | 74.5 | 82.5 | 81.5/89.3 | 66.0 | 78.8/41.0 | 76.8/76.0 | 85.9 | 66.5 | 78.8 |

Table 2: Experimental results on SuperGLUE dev set.

**Results** Table 2 reports results of our experiments on the GLUE and SuperGLUE benchmark. The first key observation is that the single model (SM) approach is outperformed by multi model (MM). This is a well known phenomena and therefore multi-model is generally adopted when the absolute best score is desired on every single task. The interesting result is that we are able to come rather close to the performance of single model with our approach. As a result, the multi-model has **16x** more parameters. To fit both GLUE and SuperGLUE, this would require **16x** the parameters. Given that our goal is to bridge the performance of a single model versus multiple models for multiple tasks, we find that this result is considerably successful. Moreover, we observe that our single-model approach outperforms the multi-model baseline by +0.6% on average across 8 tasks. We observe similar trends as on the GLUE benchmark. Naturally, the best model is the multi-model model which involves finetuning a specialized model for each task. The gap between multi-model and single-model is at 74.8 versus 73.6. Our approach bridges this gap, improving the single-model score to 74.5, competitive with the multi-model approach.

### 3.1.2 PERFORMANCE GAINS ACROSS MODEL SIZES

We investigate the gains of the proposed HyperGrid over the base model on various sizes of the T5 model. For models larger than Base, we train with 64 TPU V3 chips for $200K$ steps and select the best checkpoint for all tasks based on the benchmark score.

| Model | GLUE | SuperGLUE | AVG |
|---|---|---|---|
| Base | 84.99 | 73.55 | 79.27 |
| Ours | 85.22 (+0.27%) | 75.30 (+2.7%) | 80.26 (+1.3%) |
| Large | 88.22 | 80.04 | 84.13 |
| Ours | 88.31 (+0.1%) | 81.56 (+1.9%) | 84.94 (+1.0%) |
| 3B | 89.53 | 84.22 | 86.87 |
| Ours | **89.67** (+0.2%) | **85.75** (+1.8%) | **87.71** (+1.0%) |

Table 3: Effect of HyperGrid Transformers across all model sizes. HyperGrid improves single-model co-training consistently overly different model sizes. Improvement over SuperGLUE is greater than GLUE.

**Findings** Table 3 reports results of GLUE and SuperGLUE scores (and their macro-average). We find that performance gains on SuperGLUE averages is reasonably good (+1.9% on Large). The model still outperforms the vanilla model on GLUE with marginal performance gains. Overall, on a macro-average of 18 tasks, we find an overall +1.0% improvement across three sizes. These results show that performance gains scale with model size.

### 3.1.3 EFFECT OF MODELING CHOICES

To ascertain the effectiveness of our approach, we test different architectural variants of HyperGrid Transformers, along with other architectural variants considered during model development.

**Setup** We evaluate all four model variants of HyperGrid Transformers ($L$, $L^2$, $GL$ and $LG$). For the other architectural variants, we were mainly interested to know if a hypernetwork setup (weight gating) is better than gating on the output representations (details can be found in the supplementary material). For the base setting, we ran the baseline T5 model (single-model) four times and reported the mean and standard deviation of the runs. When comparing the performance gain of our method, we compare against the **max** run of the baseline runs. We report relative performance gains/loss against this max baseline score. We conduct ablation studies on the four composition types on the large models[1].

| Variant | GLUE | SuperGLUE | AVG |
|---|---|---|---|
| **Base Models** | | | |
| Baseline | 85.03 ($\pm$ 0.087) | 73.77 ($\pm$0.150) | 79.40 ($\pm$0.091) |
| Base (Max) | 85.11 | 73.83 | 79.40 |
| $L$ | 85.60 (+0.6%) | 74.50 (+0.9%) | 80.05 (+0.8%) |
| $L^2$ | 85.22 (+0.1%) | 75.30 (+2.0%) | 80.26 (+1.1%) |
| $GL$ | 85.12 (+0.0%) | 75.00 (+1.6%) | 80.05 (+0.8%) |
| $LG$ | 85.43 (+0.4%) | 74.78 (+1.3%) | 80.10 (+0.9%) |
| $OG$ | 85.13 (+0.0%) | 73.31 (-0.7%) | 79.22 (-0.2%) |
| OG (16) | 84.94 (-0.2%) | 73.10 (-1.0%) | 79.01 (-0.5%) |
| OG (32) | 84.84 (-0.3%) | 72.93 (-1.2%) | 78.89 (-0.6%) |
| OG (64) | 85.07 (-0.0%) | 74.11 (+0.4%) | 79.59 (+0.2%) |
| **Large Models** | | | |
| Baseline | 88.22 | 80.04 | 84.13 |
| $L$ | 88.07 (-0.2%) | 80.51 (+0.6%) | 84.29 (+0.2%) |
| $L^2$ | 88.05 (-0.2%) | 80.68 (+0.8%) | 84.36 (+0.3%) |
| $GL$ | **88.33** (+0.1%) | 80.30 (+0.3%) | 84.32 (+0.2%) |
| $LG$ | 88.31 (+0.1%) | **81.56** (+1.9%) | **84.94** (+1.0%) |

Table 4: Ablation Study. OG stands for Output Gating.

**Findings of HyperGrid Variants** Table 4 reports our key ablation results. Pertaining to results of the base models, our overall finding is that HyperGrid generally improves performance over the max baseline. Gains are mainly on SuperGLUE while maintaining good performance on GLUE. The overall average gain is about $+1\%$. Amongst the different variants of HyperGrid, the best performing model on this setup is the $L^2$ setup. On the large setting, we find that the $LG$ model performs the best while the $L$ and $L^2$ variants perform similar to the baseline.

**Is Output Gating Better?** The other architectural variants (OutGate) do not perform well and generally perform with a net loss in performance as compared to the baseline. As such, we ascertain that gating on weights is more effective than gating on the output representations. This verifies that our hypernetwork-based approach is indeed effective as opposed to simple task-conditioned output gating.

### 3.1.4 EFFECT OF GRID SIZE ON PERFORMANCE

We investigate the effect of Grid size (fan-in and fan-out) of our proposed HyperGrid method. The purpose of this experiment is to discover how fine-grained or coarse-grained the hypernetwork should be. Notably, smaller values of $d_r, d_c$ signify a more coarse-grained control of the Transformer weights.

**Setup** We searched $d_r$ (fan-in) and $d_c$ (fan-out) in the ranges of $\{4, 8, 16, 32, 128, 256\}$ and $\{8, 16, 32, 128, 256\}$ respectively and report the results on GLUE + SuperGLUE (macro-average) by varying a single value. When varying $d_r$, we took the average of all $d_c$ runs and plot the max, mean and min. Likewise, when varying $d_c$, we took the average of all $d_r$ runs and plot max, mean and average. We report scores across the $L^2$, $LG$, and $GL$ variants of HyperGrid.

**Findings pertaining to Grid Size** Figures 3 to Figures 8 illustrates performance across varied grid sizes. We observe that a clear trend exists. For most settings, a small fan-out ($d_c$) works well (e.g., 32) as noted by many spikes around this region. For fan-in ($d_r$) a smaller value also works well. However, performance gets better at higher fan-out $d_c$ values again (e.g., $> 128$). Trends are quite consistent across all three variations that we considered. These results suggest that a more coarse grid may be more effective, as the regions within the grid become larger.

---

[1]Due to the relative increased cost of searching large models, we performed a sparingly low number of ablations on large models.

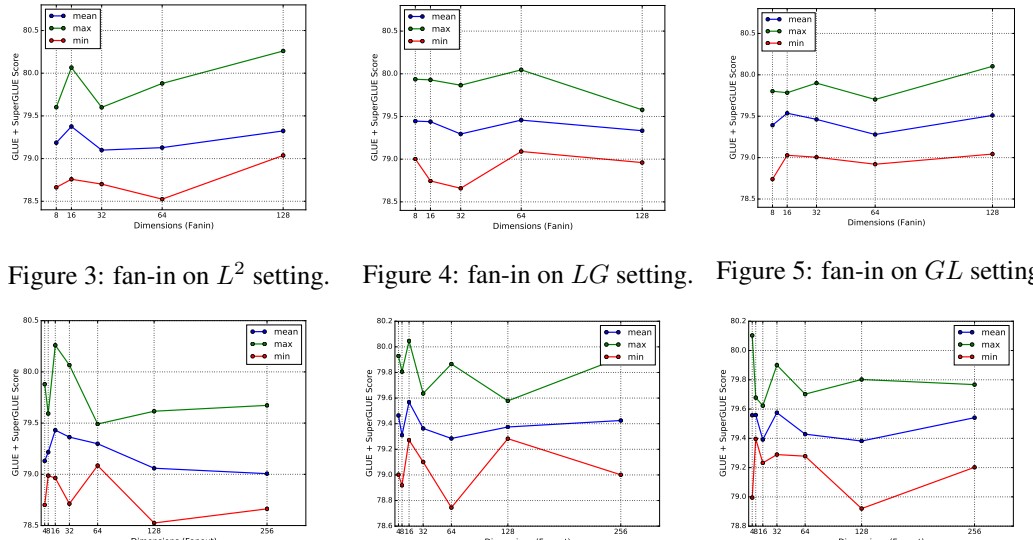

Figure 3: fan-in on $L^2$ setting.   Figure 4: fan-in on $LG$ setting.   Figure 5: fan-in on $GL$ setting.

Figure 6: fan-out on $L^2$ setting.  Figure 7: fan-out on $LG$ setting.  Figure 8: fan-out on $GL$ setting.

### 3.1.5 PERFORMANCE ON TEST SET

For our final runs, we submit our model predictions to the GLUE and SuperGLUE test servers.

| Model | $|\theta|$ | Avg | CoLA | SST | MR | STS | QQP | MNLI | QNLI | RTE | WNLI |
|---|---|---|---|---|---|---|---|---|---|---|---|
| Bert[*] | - | 80.5 | 60.5 | 94.9 | 84.5 | 86.5 | 89.3 | 86.7 | 92.7 | 70.1 | 65.1 |
| RoBERTa[*] | - | 88.1 | 67.8 | 96.7 | 89.8 | 91.9 | 90.2 | 90.8 | 95.4 | 88.2 | 89.0 |
| Albert[*] | - | - | 69.1 | 97.1 | 91.2 | 92.0 | 90.5 | 91.3 | - | 89.2 | 89.0 |
| XLNet[*] | - | - | 70.2 | 97.1 | 90.5 | 92.6 | 90.4 | 90.9 | - | 88.5 | 89.1 |
| Electra[*] | 5B | 89.4 | 71.7 | 97.1 | 90.7 | 92.5 | 90.8 | 91.3 | 95.8 | 88.5 | 92.5 |
| T5 (3B) | 48B | 88.5 | 67.1 | 97.4 | 90.0 | 89.8 | 82.1 | 91.3 | 96.3 | 91.1 | 89.7 |
| T5 (11B) | 176B | 89.7 | 70.8 | 97.1 | 90.0 | 92.1 | 82.5 | 90.9 | 96.7 | 92.5 | 93.2 |
| HGT (3B) | 3B | 88.2 | 65.6 | 97.5 | 89.0 | 91.6 | 81.9 | 90.9 | 95.9 | 90.1 | 89.7 |
| HGT (11B) | 11B | 89.4 | 69.0 | 97.6 | 89.2 | 92.6 | 82.0 | 91.3 | 96.4 | 91.5 | 93.2 |

Table 5: Test set performance on GLUE (Wang et al., 2018). Models with ∗ are large ensembles. All models are single-tasked fine-tuned except ours. Parameter costs are reported considering ensembles and cost required to fit all of GLUE and SuperGLUE.

| Model | $|\theta|$ | Avg | BQ | CB | CP | MultiRC | Record | RTE | WiC | WSC |
|---|---|---|---|---|---|---|---|---|---|---|
| BERT++ | 2.7B | 71.5 | 79.0 | 84.8/90.4 | 73.8 | 70.0/24.1 | 72.0/71.3 | 79.0 | 69.6 | 64.4 |
| RoBERTa | 56B | 84.6 | 87.1 | 90.5/95.2 | 90.6 | 84.5/52.5 | 90.6/90.0 | 88.2 | 69.9 | 89.0 |
| T5 (3B) | 48B | 86.4 | 89.9 | 90.3/94.4 | 92.0 | 86.8/58.3 | 91.2/90.4 | 90.7 | 72.1 | 90.4 |
| T5 (11B) | 176B | **88.9** | 91.0 | 93.0/96.4 | 94.8 | 88.2/62.3 | 93.3/92.5 | 92.5 | 76.1 | 93.8 |
| HGT 3B | 3B | 84.7 | 89.2 | 81.7/90.4 | 89.6 | 86.6/58.7 | 91.1/90.3 | 90.8 | 70.6 | 87.7 |
| HGT 11B | 11B | **88.9** | 91.1 | 93.9/96.8 | 94.6 | 88.1/62.4 | 93.5/92.7 | 92.3 | 74.6 | 91.6 |

Table 6: Test set performance on SuperGLUE (Wang et al., 2019). Our HyperGrid Transformers achieves competitive performance to the state-of-the-art with a single model. Parameter costs refers to total number of parameters used to fit all GLUE and SuperGLUE tasks

**Setup**    We run experiments with a 3B and 11B HyperGrid Transformer model in multi-task[2] setup (GLUE + SuperGLUE). We initialize with the T5 pre-trained checkpoints. Since this is a relatively expensive run, we only train the single model HyperGrid once using a $32 \times 128$ grid with the $LG$ (local-global) setting.  For GLUE, we compare against baselines reported in (Clark et al., 2020) which includes models such as BERT (Devlin et al., 2018), ALBERT Lan et al. (2019), RoBERTa (Liu et al., 2019b), and XLNet (Yang et al., 2019).  Note that all these models are ensembles and heavily rely on task-specific fine-tunining strategies. More details can be found in the supplementary material.

**Results on Test Set**    We find that our single model approach can achieve highly competitive results on both GLUE and SuperGLUE. Our model achieves a strong performance of $88.9$ on SuperGLUE, matching the reported T5 results while having $16$ times fewer *total* parameters.  On GLUE, the performance gap is also small, almost matching the T5 model at $89.4$ versus $89.7$. The gap on the base model remains similar at $88.2$ versus $88.5$. On SuperGLUE, our 3B model achieves $84.7$, a respectable score that matches the performance of RoBERTa ensembles fine-tuned individually with task specific tricks (Liu et al., 2019b).

## 4    RELATED WORK

Multi-task learning (MTL) (Caruana, 1997) is a long standing research problem. Learning a single unified model that does well on multiple tasks is an uphill battle given well-known problems such as catastrophic forgetting (Kirkpatrick et al., 2017).  As such, learning a large number of tasks with a single set of model parameters is an extremely challenging endeavour. Moreover, the disproportionate amount of data per task is also potentially problematic (Lee et al., 2017; Pfeiffer et al., 2020), which results in models overfitting on high resource tasks but underfitting on low resource tasks.

Early work in multi-task NLP typically considered a hierarchical taxonomy of tasks (Hashimoto et al., 2016) where a clear hierarchy of tasks exist, such as POS $\rightarrow$ Chunking $\rightarrow$ entailment.  The Joint Many-Task (JMT) model explores an incremental and hierarchical paradigm for building multi-task NLP models. Similarly, (Sanh et al., 2019) proposed a hierarchical multi-task model based on the intuition of low-level and high-level tasks.  Another line of recent work explores casting all tasks into a form of question answering problem (McCann et al., 2018) and using an interpolated pointer-generator (See et al., 2017) mechanism for generating 'answers'.

Exploiting task relatedness as a means for improved model quality has been frequently explored. In relatively recent work, (Liu et al., 2019a) proposed MTDNN, a multi-task deep neural network that shares parameters between several NLP tasks.  The model achieves strong performance on the GLUE benchmark.  However, MTDNN simply leverages MTL as a form of pretraining and uses task-specific models for final evaluation. The recent T5 (Text-to-Text Transfer Transformers) model (Raffel et al., 2019) frames all NLP problems as a Seq2Seq (Sutskever et al., 2014) problem. However, the best results are again obtained by task-specific fine-tuning.

Orthogonal to other research efforts, (Clark et al., 2019b) proposed Born Again Neural Networks (BAM), a clever way to obtain a single multi-task network by knowledge distillation. (Stickland and Murray, 2019) proposed Projected Attention Layers for task-specific fine-tuning of BERT (Devlin et al., 2018).  (Zaremoodi et al., 2018) proposed Adaptive Knowledge Sharing[3] for low-resource neural machine translation.  Our work is related to the literature surrounding hypernetworks (Ha et al., 2016) which have been found to useful in areas such as continual learning (von Oswald et al., 2019).  Learning task-adaptive parameters to avoid catastrophic forgetting has also been a go-to strategy for continual learning (Yoon et al., 2019). Outside of the NLP domain, flexible parameter sharing approaches are also dominant strategies for learning multi-task models (Ma et al., 2018; 2019).

---

[2]Since we did not co-train with the WNLI dataset due to issues stated in (Raffel et al., 2019), we simply report T5 results on WNLI. To be fair, we ignore WNLI parameter counts for all baseline models.

[3]The authors of (Raffel et al., 2019) explored this approach but did not find it to be satisfactory.

## 5   Conclusion

We proposed Hypergrid Transformers, a new Transformer architecture that leverages Grid-wise Decomposable Hyper Projections (HyperGrid), a hypernetwork-based projection layer for task conditioned weight generation. We learn and fit all GLUE and SuperGLUE tasks within the same set of model parameters and achieve competitive results to the same state-of-the-art model that is specially and individually fine-tuned on each and every task. On GLUE/SuperGLUE, this efficient single-model method results in 16x fewer parameters.

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

# 6 SUPPLEMENTARY MATERIAL

## 6.1 DATASETS

### 6.1.1 GLUE

The datasets in GLUE are CoLA (Corpus of Linguistic Acceptability) (Warstadt et al., 2018), Sentiment Treebank SST-2 Socher et al. (2013), Microsoft Research Paraphrase Corpus (MRPC) (Dolan and Brockett, 2005), QQP (Quora Question Pairs) (Iyer et al., 2017), Semantic Textual Similarity Benchmark (STSB) (Cer et al., 2017), MNLI (Multi-Genre Natural Language Inference) Williams et al. (2018), QNLI (Rajpurkar et al., 2016), RTE (Dagan et al., 2005), Winograd Schema Challenge WNLI (Levesque et al., 2012). More details can be found at `https://github.com/tensorflow/datasets/blob/master/docs/catalog/glue.md`.

### 6.1.2 SUPERGLUE

The datasets in SuperGLUE (Wang et al., 2019) are BoolQ (Boolean Questions) (Clark et al., 2019a), CB (Commitment Bank) (De Marneff et al., 2019), CoPA (Roemmele et al., 2011) (Choice of Plausible Alternatives), MultiRC (Multi-Sentence Reading Comprehension Dataset) (Khashabi et al., 2018), Record (Reading Comprehension with Commonsense Reasoning) (Zhang et al., 2018), RTE (Recognizing Textual Entailment) (Dagan et al., 2005; Bar-Haim et al., 2006; Giampiccolo et al., 2007; Bentivogli et al., 2009), Word-in-Context (WiC) (Pilehvar and os'e Camacho-Collados, 2018), and WSC (Winograd Schema Challenge) (Levesque et al., 2012). We use Tensorflow datasets for loading and preprocessing these datasets. More details can be found at `https://github.com/tensorflow/datasets/blob/master/docs/catalog/super_glue.md`.

## 6.2 EXPERIMENT SETTINGS

This section describes most of the hyperparameter settings for our experiments.

## 6.3 DATASETS AND EXPERIMENTAL SETUP

Our experiments are built upon the existing state-of-the-art model, T5. We run most of our experiments using the base T5 setting, which is comprised of $220M$ parameters. We fine-tune for a maximum of $100K$ steps. We initialize our models with the released pretrained checkpoints[4]. Our implementation is in Mesh Tensorflow (Shazeer et al., 2018). We consider the following setups for the baseline T5 model. First, we compare with the T5 results reported in the original[5] paper (Raffel et al., 2019). These results are denoted with T5$^{\dagger}$. Second, we compare with T5 (PTFT), which stands for pretrain-finetune. In this setup, we fine-tune a T5 model for each task individually following common practice. Finally, we compare with T5 (MTL) which is a fair comparison of T5 without HyperGrid. In this setting, T5 is co-trained and results are reported from a single model checkpoint selected from the best overall GLUE dev score. Note that in the MTL setting, we co-train GLUE and SuperGLUE within the same model.

**Experiments for Base Models** For all experiments with base models, we train models for $100K$ steps with a batch size of $128$. We use the *en_mix* mixture which samples each task proportionately to the number of examples in the dataset. Learning rate is a constant $0.001$ with Adafactor (Shazeer and Stern, 2018). All results for baselines are reported with scores at the last checkpoint. During fine-tuning, the embeddings are not fine-tuned. Experiments are run with 16 TPU V3 chips and are typically completed in about $8$ to $10$ hours.

**Experiments with Large Models** We increased the search for large models to $200K$ steps pick the best checkpoint for all models based on the best GLUE score. Experiment and hyperparameter settings remain identical although we use $64$ TPU V3 chips for finetuning which typically take about $12$ hours to complete.

---

[4]`https://github.com/google-research/text-to-text-transfer-transformer`.

[5]This model is not directly comparable as they used less pretraining steps. No dev score results on a comparable setup is reported. We report this score for the sake of completeness.

**Experiments with 3B and 11B Models**    For the large models, we only use $1 - 2$ HyperGrid configurations 32x128 or 32x256 in $LG$ mode for finetuning the model. We submit each model only once to the leaderboard[6]. Finetuning hyperparameters remain identical. We pick a single checkpoint based on the best GLUE score. Finetuning for the $3B$ model is using 64 TPU V3 chips and the $11B$ model is fine-tuned with 128 TPU V3 chips.

## 6.4    COMPARING WITH OUTPUT GATING

One of the model architecture variants we compared with is Output Gating. It can be formulated as:

$$Y = \max(\boldsymbol{W}x + b, 0) \odot (\sigma(\boldsymbol{U}X)\mathbf{1}^\top) \tag{8}$$

Comparing to the HyperGrid, which gates the weights in the Relu layer, output gating directly gates the Relu layer outputs. We can apply either the basic projection method (Equation (2)), or the grid-wise projection method with block-wise projection on layer outputs.

There are two key differences: (1) Output Gating applies sigmoid gating on Relu layer outputs, while HyperGrid applies sigmoid gating on weights before the Relu function. Output gating is similar to the Mixture-of-Expert architecture while concatenating the expert outputs. (2) Based on this formulation, the full grid-based projection cannot be applied to output gating.

## 6.5    FURTHER ARCHITECTURAL ABLATIONS

We include more architectural ablations to supplement the results and findings of the paper.

**Ablation Architectures**    We run experiments for 6 different ablations. The first (1) - (4) is concerning with where to apply HyperGrid. (1) applies on the entire network (all QKV + both FFNs), (2) applies it to both positional FFNs, (3) applies it on the 1st FFN only and (4) applies it on the 2nd FFN only. For (5) and (6), we evaluate on different weighting schemes for HyperGrid. (5) projects HyperGrid to a scalar value by pooling (in essense, instead of upsampling the grid, we downsample the grid into a scalar value). As fo (6), this is the setting where the the row and column size of the HyperGrid is essentially the same as the FFN.

| Ablation | AVG |
|---|---|
| Baseline | 79.40 |
| (1) All (QKV Transforms + FFN) | 79.05 |
| (2) FFN (both FFN) | 80.20 |
| (3) FFN (1st FFN only) | 80.23 |
| (4) FFN (2nd FNN only) | **80.26** |
| (5) Scalar HyperGrid | 79.20 |
| (6) Max-Grid HyperGrid | 80.10 |

Table 7: More results (SuperGLUE + GLUE) studies on applying HyperGrid to different parts of the Transformer.

**Results**    Our ablation studies show that the best place to place the FFN is only at the 2nd FFN. Using HyperGrid on all the layers, apart from slowing down the network, would also degrade performance. This can be observed because the difference between (1) and (2) is the addition of QKV HyperGrids to the model. In (5) and (6), we also note that Scalar HyperGrid performs poorly and worse than the Baseline model. Finally, the Max-Grid HyperGrid performs reasonably well. However, this incurs a huge cost because the hyper network is now larger (as composed to learning small grids and upsampling).

---

[6]Discounting submissions that turn out to be incomplete or error submissions.

