# OpenReview forum: "HyperGrid Transformers: Towards A Single Model for Multiple Tasks"
_ICLR.cc/2021/Conference — ICLR 2021 Poster_

### Official Review · AnonReviewer4 · 2020-10-28
**Review #4**

**Rating:** 6
**Confidence:** 3

**Review:**

The authors propose HyperGrid Transformers with a decomposable hypernet-work that learns grid-wise projections to specialize regions in weight matrices for different tasks. Usually, people would use different models to solve different tasks respectively. In this paper, the authors focus on using a single model to solve all tasks and it will save a lot of model parameters for natural language understanding. And the authors have done comprehensive experiments on GLUE and SuperGLUE, and prove that the proposed single model can achieve much better performance than baseline and competitive performance with multiple task-specific models.

Pros:
1. The idea to make use of decomposable grid wise projection is interesting. This is to somehow add regularization to the weights.
2. The proposed method has been widely evaluated on GLUE/SuperGLUE tasks, and achieve good performance.

Cons:
1. The baseline details are not clear. When using a single model as baseline, how many layers are shared across tasks? What's the sample strategy for different tasks? Is it possible to train a single model on multi-tasks for some steps, then fix most layers and only finetune some task specific layers? I feel the baseline is a bit weak, although I cannot come up with a stronger one that can be easily adapted to the pertained model.
2. It seems "Task Conditioning" is a very important trick. The authors should have some ablation study on it. Or maybe add it to the baseline.

Overall, it's great to see people working a single model to solve all tasks. And I would be happy to increase my score if the authors could convince me regarding the baseline which is quite tricky.

####update####
The experiment results are not surprised, but strong enough. Still no very strong baseline provided in this submission, but it might be good to set up a benchmark in this direction. However, T5 model needs more computational resource and the experiment results are hard to replicate. Overall, I would like to keep rating.

---

> ### Author Response · Authors · 2020-11-23
> **Response to Review**
>
> Thanks for the insightful comments and feedback, along with taking the time to review our paper. We are happy and excited that you find our work interesting.
>
> For the single model baseline, this is the identical T5 model but *without* the hypergrid layers. This follows the T5 setup where they train all the models within one single model, i.e., all layers and parameters are shared. In the T5 model (https://arxiv.org/abs/1910.10683), the task conditioning happens by providing the model with a task prefix which interacts with the input sample via self-attention. In our opinion, this is quite a fair baseline because this essentially compares with our model without the HyperGrid layers. This way, we are able to observe the effect of HyperGrid directly.
>
>  In our experiments, we also compared with a very strong baseline, which is the multi-model baseline which finetunes a fresh model for every task (and also picking the best checkpoint for each task). This is a very strong baseline because the model is allowed to pick the best checkpoint for each task while this is prohibited in single-model (ours) approach.
>
> As for the sampling strategy, we used a proportionate mix following the T5 model setup. This setup samples the tasks proportionately according to the actual dataset sizes of each individual task. If you look at the open source T5 code, this refers to the en_mix mixture. Notably, the sampling strategy for our method (HyperGrid) remains identical to the baseline to enable a fair apples to apples comparison.
>
> Thank you for the feedback. We have made this clearer in the revised version of the paper (under “Details of Baseline”)
>
> Regarding the suggestion of training in multi-task setting for some steps and then fine-tuning on task specific layers, we think that this is indeed a great suggestion. We will definitely try this out in the future. Thanks for the suggestion.
>
> As for task conditioning, we show in Table 4 an extensive ablation study on the different methods for task conditioning. We compare (on base and large sizes), at least 8 different variants of task conditioning. Note that the original T5 Baseline is also a form of task conditioning (by passing the task as an input prefix). We also added 6 different ways of task conditioning with HyperGrid variants in the revised version of our paper. Thanks for the suggestion!
>
> Thanks for taking the time to review our paper. Many thanks for the interesting and insightful suggestions.

---

### Official Review · AnonReviewer2 · 2020-10-29
**An empirically good method for a single multi-task Transformer**

**Rating:** 6
**Confidence:** 3

**Review:**

This manuscript presents a HyperGrid Transformer, which is engaged in learning a single model to account for multi-tasks in NLP. The core idea of HyperGrid Transformer is to learn task-conditional dynamic weights in a grid-wise manner in the feed-forward layers, where the weights are factorized in local and global components. This idea is simple, materializing the goal of reducing the parameter cost for the used multi-task network. However, the conducted experiments look nice, showing promising performance on GLUE/SuperGLUE. Therefore, from my point of view, this work is worthy of a publication at ICLR.

---

> ### Author Response · Authors · 2020-11-23
> **Response to Review**
>
> Dear Reviewer,
>
> Thanks for taking the time to review our paper, along with the great feedback and comments.
>
> We are also glad that you liked our paper and find our work worthy of a publication at ICLR!
>
> Thanks!

---

### Official Review · AnonReviewer5 · 2020-11-08
**Interesting paper about applying hypernetworks to Transformers**

**Rating:** 7
**Confidence:** 3

**Review:**

This paper presents a HyperGrid Transformer approach to fine-tuning, where one takes a pre-trained transformer model and then modifies it by introducing hypernetworks that modify the 2nd FFN in each transformer block by generating additional weights conditioned on input. These hyper-networks are trained on all tasks in GLUE/SuperGLUE datasets simultaneously and are task aware through prefixing of a task specific token to input. This allows one to fine-tune only a small number of parameters and end up with a model that performs quite well on all tasks at the same time, not much worse than fine-tuning the entire transformer model on all of these tasks.

This is an interesting paper in the area of hypernetworks with results suggesting potential for impact where one can achieve good accuracies on tasks without having to fully fine-tune gigantic pre-trained models.

I do have some questions for the authors though:

-  local vs global - global seems to be just like learning another weight matrix not conditioned on anything? from the name one would expect local to be conditioned on some specific parts of the input while global is conditioned on entire input. what is the intuition on why they help?

-  what's the intuition between the differences of performance of the various setups (LG, GL, L^2, etc)

-  figure 9 doesn't exist. if coarser variants are better, what happens when the entire weight matrix is treated as 1 block (so you just learn a scalar weight)? what about learning a single scalar for each weight in the FFN (i.e. block size is 1)?

-  have you tried adding dynamic weights to projections in the multi head self attention modules (e.g. to projections for q,k,v)

-  table 1 - why are QNLI results so much worse for HGT(LG) than all other results, but seems to be doing better on most other tasks; how stable are all of your results (i.e. what is the variance across seeds?)

- parameter counts are confusing - how does one compute that a "multiple networks" needs 3.2b params, but a "single network" needs 0.2b params? are these the trainable weights for each setup, so you count 16 tasks x 0.2b weights? maybe it is better to report total num of params + trained params or else somehow make it more clear what the number of parameters means

- have you tried finetuning the hypernetworks on individual tasks?

---- update:
Thanks for the update. I guess the "intuition" is driven mostly by empirical results, which I suppose is ok but may be worth digging into a bit more. I have updated my rating.

---

> ### Author Response · Authors · 2020-11-23
> **Response to Review**
>
> Dear Reviewer,
>
> Thanks for insightful comments and feedback! We are excited that you find our work interesting!
>
> We are happy to answer your detailed comments and questions.
>
> Regarding intuition of various variants, our results show that local only variants tend to do better on the base size but global variants LG and GL work better when model size gets larger. This is why we use global-local variants in the final sweeps for the extremely large models given limited compute. The difference of Transformer FFN size directly affects HyperGrid since Hypergrid is applied directly to these FFN layers. We believe that G might have a regularizing effect which helps larger models.
>
> Regarding local vs global, this is indeed a great question. The global vector is not conditioned on anything and is treated as trained parameters with the rest of the network. The local vector is conditioned on the input sequence (and consequently, the entire input because of self-attention).
>
> Regarding Figure 9, thanks for spotting that! We have fixed that in the revision!
>
> Regarding adding dynamic weights to the projections for QKV, we have added new (ablation) results for this to the supplementary material .Here the baseline model does 79.40 while All layers (QKV + FFN) achieves 79.04. Applying on the 2nd FFN only achieves 80.26 and 1st FFN only 80.23. Both FFN does 80.20. Thanks for the suggestion!
>
> Regarding the scalar and max-grid blocks, we have added these ablations too to the “Further Architectural Ablations” section in the supplementary material. Thanks for these suggestions! Regarding coarser variants compared to very fine-grained grids, these generally do well but incur costs and have diminishing effects after a certain point (this is why we searched up to 256 for d_r and d_c in our paper). In our initial experiments, we went further to 512 and 1024 and beyond but did not see any considerable gains.
>
> Regarding the parameter counts, these are the total number of parameters to serve all the tasks (setup). Assuming each single model has 0.2B trainable parameters, the cost of performing the entire suite of tasks (16 tasks) would be 3.2B (trainable parameters). Since in the multi-model approach, each single model is finetuned on 16x downstream tasks and therefore would require 16x the parameters in total to serve all 16 tasks. Thanks for the suggestions! We have added a note that $|\theta|$ in the papers refers to the total number of parameters for the setup (this is right at the start of the experiment section).
>
> Regarding variance, generally the variance of aggregate scores (superglue or glue) is pretty low (less than 0.1 in SuperGLUE + GLUE score across multiple (>5) runs). The model learns to make trade-offs between different tasks which may explain the task variance.
>
> In early experiments, in tried HyperGrid Transformers on a select subset (e.g., MRPC, SST etc) of tasks individually. We were not able to observe any significant gain in performance. There are possibly intuitive reasons for this - since HyperGrid is a method for controlling parameters in a task-adaptive way, when the number of tasks=1, this is just simply adding a gating weights to the FFN. We feel that the strength of HyperGrid is primarily designed for coping with multiple tasks.
>
> Thanks for taking time to review our paper, along with the interesting and insightful feedback.

---

### Author Response · Authors · 2020-11-24
**Summary of Updates in Revised Paper**

Thanks to all reviewers and area chairs for the hard work!

Please find the following summary of updates and changes to our paper:

1. We added a section "Details about baselines" to make the baseline details and setups clearer.
2. We added new architectural ablations (6 new results, including the ones requested by Reviewer5) in the supplementary material.
3. We added details on how params are counted in our tables.

Thank you for all for the insightful review and helping to improve our paper!

---

### Decision · Program_Chairs · 2021-01-07
**Final Decision**

**Decision:**

Accept (Poster)

**Comment:**

The paper proposes "HyperGrid Transformers" a modified transformer architecture for learning a single model for multiple tasks in NLP.  The proposed method was evaluated on popular GLUE/SuperGLUE tasks and reported competitive results with the baselines (the improvements are somewhat marginal). The paper contains some interesting idea of using a decomposable hypernetwork to learn grid-wise projections for different tasks, which may not be particularly novel in machine learning context but seems new for multitask NLP. Reviewers generally agree the paper is above acceptance bar, however some concerns were raised about clarity of baselines and fairness of experimental comparison as well as stronger baselines. Authors improved some of them in the rebuttal, but there is still some room to further improve the quality of presentation and writing in the final version.